# Direct selection of functional fluorescent-protein antibody fusions by yeast display

**Nileena Velappan**[1][☉]*, **Fortunato Ferrara**[2][☉], **Sara D'Angelo**[2], **Devin Close**[3], **Leslie Naranjo**[2], **Madeline R. Bolding**[1], **Sarah C. Mozden**[1], **Camille B. Troup**[4], **Donna K. McCullough**[5], **Analyssa Gomez**[1], **Marijo Kedge**[1], **Andrew R. M. Bradbury**[2]*

**1** Bioscience Division, Los Alamos National Laboratory, Los Alamos, NM, United States of America, **2** Specifica Inc., Santa Fe, NM, United States of America, **3** Arup Laboratories, Salt Lake City, UT, United States of America, **4** Carterra Inc., Salt Lake City, UT, United States of America, **5** Microbiology Department, University of Tennessee, Knoxville, TN, United States of America

☉ These authors contributed equally to this work.

\* abradbury@specifica.bio (ARMB); nileena@lanl.gov (NV)

**Data Availability Statement:** All relevant data are within the paper and its Supporting Information files.

## Abstract

Antibodies are important reagents for research, diagnostics, and therapeutics. Many examples of chimeric proteins combining the specific target recognition of antibodies with complementing functionalities such as fluorescence, toxicity or enzymatic activity have been described. However, antibodies selected solely on the basis of their binding specificities are not necessarily ideal candidates for the construction of chimeras. Here, we describe a high throughput method based on yeast display to directly select antibodies most suitable for conversion to fluorescent chimera. A library of scFv binders was converted to a fluorescent chimeric form, by cloning thermal green protein into the linker between VH and VL, and directly selecting for both binding and fluorescent functionality. This allowed us to directly identify antibodies functional in the single chain TGP format, that manifest higher protein expression, easier protein purification, and one-step binding assays.

## Introduction

Protein chimeras are recombinant proteins created by linking the genes of different proteins, with the intention of combining their functional properties [1]. These fusion proteins often combine the binding activity of one protein with the enzymatic activity, toxicity, fluorescence, improved solubility, expression or folding of a second protein. Antibodies are most used to confer binding activity, and scFvs the most commonly used format for antibody chimeras. These reprise the antigen-binding properties of full-length antibodies in a smaller, single gene construct [2–4] and are usually selected from immune or naïve libraries using phage display [5–12]. The properties of selected antibodies can be significantly enhanced by combining phage and yeast display, which provides greater control over selection parameters and improves the affinity and diversity of selected scFvs [13–17].

Engineered fluorescent proteins play major roles in biological research and are extensively used to study gene expression, protein function, tracking, and protein-protein interaction [18]. Chimeras in which antibody fragments are combined with fluorescent proteins provide

**Funding:** This work was supported by National Institutes of Health Grant P50GM085273 Foundation for the National Institutes of Health [P50GM085273]; Los Alamos National Laboratory's Laboratory Directed Research & Development grant 20220108ER. Specifica Inc, a Q2 Solutions Company, provided support in the form of salaries for authors FF, SD, LN and ARMB, but did not have any additional role in the study design, data collection and analysis, decision to publish, or preparation of the manuscript. The specific roles of these authors are articulated in the 'author contributions' section.

**Competing interests:** FF, SD, LN and ARMB are employed at Specifica Inc, a Q2 Solutions Company. CBT is employed at Carterra. The commercial affiliations do not alter our adherence to PLOS ONE policies on sharing data and materials. The other authors declare no conflict of interest.

advantages during the evaluation of protein expression, folding, purification and the detection of binding. While antibodies have been combined with fluorescent proteins in many different formats [19–26], there is an inherent complexity involved in the creation of such chimeras since antibodies are secreted proteins and fluorescent proteins are expressed in the cytoplasm. Consequently, many published chimeras have very low expression levels [20], a problem that has been addressed with the split GFP system [27], in which a short tag derived from GFP, rather than GFP itself, is fused to a scFv [28], or by using particularly stable scFvs [22] or nanobodies [29]. An unexpected partial solution to this problem was the finding that fluorescent proteins could be used as linkers between VH and VL [30,31], with the resultant chimeras often better expressed than the parental scFv [32]. We named these chimeric proteins scFPs (single chain fluorescent proteins) [32], according to the fluorescent protein used–scTGP, when thermal green protein [33] was used, and scGFP, when superfolder GFP [34] was used. Unfortunately, while some poorly expressed scFvs were expressed far better as scFPs, not all scFPs were equally well-expressed. As it was impossible to predict *a priori* which scFvs would have their expressions improved, we converted highly specific scFvs to scTGPs and scGFPs, one at a time, to evaluate their functionality. When expressed well, the intrinsically fluorescent scFPs were effectively displayed on yeast and functioned similarly to the corresponding scFvs with equivalent affinities. Their intrinsic fluorescence allowed straightforward assessment of their expression levels and purification. scFPs do not necessarily need to be secreted and can sometimes be effectively expressed in the bacterial cytoplasm, sometimes at higher levels than the corresponding scFv.

Independently of the library used [16,35], we routinely combine phage with yeast display, yielding one to thousands of different binders [14–17,36–38]. scFv affinity and the degree of scFv diversity depends upon the library used. As not all selected scFvs can be effectively converted to scFPs, we explored the possibility of establishing a high throughput selection strategy that would directly select scFvs that were also functional as scFPs. This involved selecting for intrinsic fluorescence and binding—the two desirable qualities inherent to a functional scFP–simultaneously during yeast display flow cytometry, to directly sort functional scTGPs without the need to test them individually. This concept was validated using a sub-library of scFvs selected against two human enzymes: Cyclin-Dependent Kinase 2A (CDK2A), a human protein important in cancer biology, and Ubiquitin Specific Peptidase 11 (USP11), which is associated with bronchiectasis [15]. We discuss the advantages of this high throughput selection method, the utility of the scTGPs in various techniques and the possibility of utilizing similar methodology for the development of other novel chimeric proteins.

## Materials and methods

### Construction of single scTGPs

The scTGP clones p1-1[10], 14C2 [12,39] and Z3 [12] were constructed using circular polymerase extension cloning (CPEC) assembly [40] from yeast display (pDNL6) clones that contained the corresponding scFv gene [32]. Briefly, the scFv genes and the vector sequences were amplified by inverse PCR and the TGP sequence was amplified using standard PCR with primers containing overlapping sequences that facilitated CPEC assembly. The assembled plasmids were transformed into Omnimax T1 cells (ThermoFisher, C854003) and sequenced prior to transformation into *S. cerevisiae* EBY100 yeast cells (ATCC, MYA-4941) using YEAST1Kit (Sigma Inc, Yeast1). For bacterial expression of scTGPs, the genes were sub-cloned into pET-TEV expression plasmid system using the restriction enzymes BssHII and NheI (New England Biolabs) and sequenced prior to protein expression.

## Yeast display analysis

Yeast display analyses using scFvs and scFPs cloned into the pDNL6 yeast display vector in EBY100 cells were performed as previously described [5,10,15,32]. Here the transformed yeast cells were grown in SD/CAA media at 30˚C until $OD_{600}$ >2 and then scFv/scTGP display induced in SG/R CAA media for 36–48 hrs at 20˚C. The yeast were subsequently washed with yeast wash buffer (YWB, 30mM Tris pH 8.0 with 0.5% BSA) and allowed to bind biotinylated peptides and proteins at 100nM concentration for 30–60 min at RT with mild rotation. The antigen-bound yeast were washed twice by centrifugation and stained with anti-SV5 (ThermoFisher #37–750, 1μg/mL) conjugated with phycoerythrin (PE) using a commercial kit (Abcam #ab102918), and Alexa 633-labeled streptavidin (ThermoFisher Scientific, S21375, 5 μg/mL) for 30 minutes. The yeast samples were again washed by centrifugation and analyzed using FACS Aria and/or LSR II cytometers (Becton Dickinson). 10,000 events were collected for each analysis and mean fluorescent intensity (MFI) values was calculated; experiments were conducted in triplicate. The PE signal (antibody display) is detected at 561nm excitation with a 582/15nm emission filter. Binding of biotinylated target molecules was measured using streptavidin Alexa 633, excited at 635nm and with a 660/20 emission filter.

Affinity measurements were obtained using yeast display by measuring the amount of fluorescently labeled antigen bound to the yeast surface at different concentrations at equilibrium. The estimated $K_D$ is obtained by determining the concentration of antigen corresponding to half maximal binding. $K_D$ determination was performed using GraphPad Prism® software using the equation for one site-specific binding, which uses nonlinear regression analysis to fit the curve, as described previously [10,15,32].

## scTGP expression and purification in *E. coli*

Our previous work [32] had shown that pET vector-based cytoplasmic expression in *E.coli* is the optimal system for scTGP expression and purification. Briefly, *E. coli* BL21 transformed with scTGP clones were grown in LB media and transferred to auto induction (AI) media for protein expression [41]. Protein expression was performed at 20˚C for 48–72 hours and cell pellets collected by centrifugation. Soluble protein was isolated using POP culture reagent (EMD Millipore, #71092) by incubating cell pellets from 1-2mL of culture with 100μL of 1/10 POP culture reagent for 30 minutes followed by resuspension in Tris or PBS buffer to a final volume of 1mL. The pellets were further centrifuged at 20,000g for 20 minutes and supernatant containing soluble protein was removed for further analyses.

Ni-NTA agarose-based protein purification was performed from 100mL or 1L cultures. The bacterial pellets were resuspended in TNG buffer (50 mM Tris pH 8.0, 150 mM NaCl, 10% v/v glycerol) containing DNase 1 (ThermoFisher, #PI89836,) and EDTA-free protease inhibitor tablet (Roche Inc., #11873580001) and lysed using a cell homogenizer (Avestin Inc.). Centrifugation was used to remove cell debris and the insoluble fraction, and protein purification was performed as previously described [32].

Protein quantification was performed using two different methods—by gel electrophoresis and/or absorbance at 280 nm. All protein electrophoresis was performed using 4–20% miniprotean TGX gels (Biorad, #456–8094). Protein quantification with known quantities of sfGFP and/or BSA was performed using ImageLab software (Biorad Inc.). Fluorescence -based protein quantification again using sfGFP was performed using with excitation set at 488 nm and emission set to 530 nm using Infinite M200 spectrophotometer (Tecan Inc.).

## Phage selection and yeast sorting

ScFv antibodies were first selected by phage display from a naive library [35] using a previously described strategy [14, 15]. Briefly, selections were performed with an automated magnetic bead system (Kingfisher, ThermoFisher) using 50 nM of biotinylated CDK2A and USP11 proteins [42] incubated with the phage antibody library. Binding phages were captured with streptavidin-conjugated magnetic beads (Dynabeads M-280), and non-binding phage were removed from the beads after a series of washing steps. Bound phages were released from the beads by acid elution and used to infect F' bacteria (Ominmax-2T1, ThermoFisher). After propagation of the eluted phage the selection cycle was repeated, for a total of two rounds for each target. The second-round phage outputs were PCR amplified with specific primers that introduce 5' extensions overlapping with the yeast display vector pDNL6, allowing cloning by homologous recombination after co-transforming the vector and amplification products into competent yeast cells [14,15]. The transformed yeast cells were enriched for target specific binders with 2 rounds of fluorescence activated cell sorting (FACSAria, Becton Dickinson), following previously described protocols [14,15]. Cells were labeled with streptavidin-Alexa-Fluor 633 (1:400) to detect binding of the biotinylated target antigens and 0.5 µg/mL of anti-SV5-PE to assess scFv display levels. Yeast clones showing both antigen binding (Alexa-Fluor 633 positives) and display (PE positives) were sorted.

## Construction of scTGP libraries

The construction of scTGP libraries was conducted using a similar strategy to that applied for individual scFvs [32]: the entire scFv selection outputs obtained after the yeast display enrichment was used to prepare yeast plasmids and as the target DNA for inverse PCR using primers DCO 111 (`TCCTCAAGCGGTACCCAGGTGCA`) and DCO 112 (`TCTGGAGGGTCGACCATAA CTTCG`). CPEC assembly was performed with TGP specific primers DCO 109 (`TCTGGAGGG TCGACCATAACTTCGGTAATTAAACCGGAAATGAAAATTAAATTGCG`) and DCO 110 (`TATT CTGGCGGAGGCAGCGGATCCTCAAGCGGTACCCAGGTACA`) as described above. The CPEC assembly reaction was purified using minielute columns (Qiagen, 28006), transformed into Omnimax T1 cells and plated on Carb/glu plates (carbenicillin 50 µg/mL, 3% glucose) and incubated overnight at 37°C. The cells were scraped the following morning and plasmid DNA was prepared using Qiagen miniprep kit (Qiagen, 27106) and subsequently used for yeast transformation.

## Analysis and enrichment of the scTGP libraries

Yeast cells containing scFv outputs selected on CD2KA and USP11 and yeast transformed with the same outputs cloned as scTGP were grown and induced. Antigen staining was performed at 100 nM. Washes and secondary fluorescent reagent incubations were carried out as described above. Yeast containing outputs in the scTGP format were further enriched by yeast sorting using the green fluorescence signal to assess display levels. The sorted yeast libraries were cultured in SD/CAA media and a modified Qiagen miniprep protocol was used to prepare plasmids from yeast cells. In this protocol, yeast cell walls are disrupted by vortexing at high speed for 10 minutes with acid washed beads (Sigma Inc. #G8772) and double the amount of P1 and P2 buffers are added. The manufacturer's instructions were followed for the remainder of the protocol and the yeast plasmid DNA was used in subsequent sequencing reactions. The yeast plasmid DNA was also used to transform Omnimax T1 cells to obtain single colonies for Sanger sequencing of scTGP clones.

## Screening of scTGP monoclonal antibodies

Individual clones obtained from the sorted scTGP populations were tested by flow cytometry for their specificity towards their targets (CDK2A and USP11), using 100 nM of antigen. After confirming the specificity of binding, the scTGP genes were sub-cloned into the pETCK3 cytoplasmic protein expression vector using restriction enzymes *BssHII* and *NheI*. Sequence verified plasmids were used for protein expression in BL21 cells (NEB, #C2530H). Protein induction, purification, and quantification was performed as described previously [32]. Equal quantities of the scTGPs, based on fluorescence values, were used as input for the fluorescence linked immunosorbent assay (FLISA) assays [43], performed to confirm and compare binding activity of the scTGP molecules. Wells of black maxisorp plates (ThermoFisher #43711) were coated with 1 μg of antigen per well overnight at 4˚C. The following day, after blocking the wells with 2% BSA in PBS solution, equal amounts of purified scTGPs (1μg) diluted in 0.2% PBS-BSA, were incubated for 1 h at room temperature. After washing three times with PBST (0.5% Tween-20 in PBS) and three times with PBS to remove nonspecific interactions, the fluorescent values were read at 488 nm excitation and 530 nm emission. The experiments were performed in triplicates.

The protein expression levels of the selected scTGPs were analyzed in two different ways: i) small scale expression using 2mL of media in 96 deep well plates (ThermoFisher #951657) where soluble protein was obtained using POP culture as described above. Each construct was grown in three wells and the protein expression was quantified using the fluorescent signal normalized on known amounts of purified sfGFP. ii) protein expression using 1L cultures was performed as described earlier. Preparation of scFv proteins were also performed similarly except for the following changes: a) scFv expression was directed to the periplasm via the pelB leader present in the pEP vector system b) protein expression was induced for 48hrs (compared to 72hrs for scTGPs). Protein quantification for both scFv and scTGP clones was also performed by comparing the intensity of Coomassie stained bands following gel electrophoresis with known protein standards. Band quantification was performed using ImageLab software (Biorad Inc.)

## Next generation sequencing of scFv and scTGP sorted libraries

The plasmid DNA collected from the yeast cells of the anti-CDK2A and anti-USP11 second sort outputs obtained both as scFvs and as scTGPs were used as templates for PCR amplifications targeting the HCDR3 region of the scFvs and scTGPs. The HCDR3s were amplified with a set of forward primers carrying one of the Ion Torrent sequencing adaptors and mapping to the framework region upstream of HCDR3 in combination with a barcoded reverse primer mapping to the SV5 tag region of the yeast display vector and carrying the second adaptor necessary for sequencing. Primer sequences and detailed method are described by D'Angelo et al [37]. After amplification with the proofreading Phusion polymerase (NEB), gel extraction, and quantification (Q-bit, HS-DNA kit, Invitrogen), the sequencing libraries were processed using the Ion Xpress Amplicon library protocol and then prepared for sequencing on the Ion 316 Chip (Life Technologies). The analysis of the sequences was performed using the AbMining Toolbox as described by D'Angelo et al [37].

## Fluorescence microscopy

The human embryonic kidney (HEK) 293 cell line transfected with the M2 influenza protein and the corresponding HEK 293 untransfected control cell line was kindly supplied by Dr. Mark Tompkins (University of Georgia). Cell propagation and M2 induction were performed as described in Gabbard et al [39]. For microscopy experiments, cells were seeded onto

8-chambered borosilicate glass slides (Nunc LabTek, #155411), fixed with 4% paraformalde-hyde (PFA) in 1xPBS for 15 minutes and washed twice with 1xPBS. The wells were blocked with 2% BSA for 1hr. 1μg of scTGP 14C2 and scTGP Z3 was added to three wells of HEK 293 M2 and HEK 293 cells and incubated overnight at 4˚C. Unbound reagents were washed with 3X PBST and 3X PBS. Fluorescent microscopy images were taken at 488nm excitation and 530nm emission for both cell types with both scTGPs. Fluorescent images were obtained using Zeiss Axio Observer Z.1.

## Large scale expression of scFv and scTGP clones and size exclusion chromatography (SEC) analysis

The DNA sequences for the CDK2A and USP11 gene-synthetized clones expressed both as scFvs and scTGPs are shown in S1 Table. Protein production for the four scFvs and corre-sponding scTGPs was performed in 500 mL of AI media followed by protein purification using Talon affinity resin as described above. The eluted fraction was further purified by TEV protease (1mg/mL) digestion. Here, the eluted protein was dialyzed in TEV buffer (20 mM Tris pH 8.0, 100 mM NaCl, 0.5 mM EDTA, 3 mM reduced glutathione). 200 mL of TEV prote-ase was added to 1800 mL of eluted fraction and dialyzed overnight at RT. The following morning, uncut proteins and TEV protease were captured on Ni-Agarose beads and the super-natant collected as the TEV purified protein. The quality of proteins was assessed by SDS PAGE. Size exclusion chromatography (SEC) for scTGPs were performed using HiLoad Superdex 16/600 75 pg column and the AKTAprime pure protein purification system (GE healthcare) according to the manufacturer's instructions. Fractions with the highest UV read-ings and fluorescence values were collected and the quality of the proteins was assessed by SDS PAGE. The molecular weight was calculated based on standard calibration curves derived from protein standards purchased from Biorad (Catalog # 1511901), using the equation $Kav = (Ve—Vo)/(Vt—Vo)$ where $Ve$ = elution volume, $Vo$ = column void volume (44.07 mL based on Blue dextran elution volume), $Vt$ = total bed volume (120 mL).

## Kinetic analysis by SPR

A covalent array using amine coupling was prepared on a Carterra LSA by immobilizing 4 scTGPs (A8, F2, B6 and C3) (ligands) to an HC30M sensor chip (30 nm linear polycarboxylate surface Carterra #4279). Immobilization running buffer was 25 mM MES (2-(N-morpholino) ethanesulfonic acid) pH 5.5 with 150 mM NaCl and 0.05% Tween 20. The sensor chip was conditioned with serial 1-minute injections 50 mM NaOH, 1M NaCl and 10 mM Glycine, pH 2.0. Immobilization of scTGPs was performed using the array prep wizard. The sensor surface was activated with 133 mM EDC (1-ethyl-3-(3-dimethylaminopropyl)carbodiimide hydro-chloride (ThermoFisher #PG82079) and 33 mM S-NHS (Sulfo-NHS (N-hydroxysulfosuccini-mide (ThermoFisher #24510) in 100 mM MES pH 5.5 for 8 minutes using the single flow cell (SFC). The scTGP ligands were immobilized in triplicate in a 6 point, three-fold dilution series starting at 10 mM and down to 40 nM in 10 mM NaAcetate buffer pH 4.25 + 0.05% Tween 20 for 15 minutes, with each sample immobilized twice on the array. Active groups were quenched with 1M Ethanolamine pH 8.5 for 5 minutes. Kinetic analysis was performed at 25˚C using 1X HBSTE (10 mM 4-(2-hydroxyethyl)-1-piperazineethanesulfonic acid (HEPES), 150 mM NaCl, 3 mM EDTA, and 0.05% Tween 20) with 0.5 mg/mL bovine serum albumin (BSA). As the running buffer and sample diluent. Analytes (CDK2A or USP11) were injected for 5 minutes with a 10-minute dissociation. In serial injections, buffer blanks were injected followed by analyte injections from the lowest to highest concentration (6 point, 3 fold titra-tion series ranging from 2 nM to 500 nM) of each analyte with no regeneration. Kinetics data

analysis was performed using Carterra Kinetics software, v. 1.8.0.3603. Data were double referenced by subtracting an unmodified reference location and a leading buffer blank cycle. Data was y-aligned to the common baseline before the lowest analyte concentration and globally fit using a 1:1 Langmuir model to determine the association rate constant $k_a$ and dissociation rate constant $K_D$ as well as the affinity constant $K_D$ ($k_d$/ $k_a$). Sensorgrams from appropriate ligand densities which had $R_{max}$ values between 50 and 200 were included and treated as replicates in the reported rate values.

## Results

### Challenges associated with converting individual scFv to scTGPs

In previous work [32] three different scFvs were successfully converted to scTGPs. Subsequently, we attempted the conversion of scFvs Z3 and 14C2, derived from two antibodies recognizing the external 24 amino acid domain of the Matrix 2 (M2) protein of the influenza A virus [12,39]. Yeast display-based analyses, and relative $K_D$ measurements, obtained from flow titration curved (Fig 1A), showed nearly equivalent values for the scFv and scTGP versions of 14C2 and Z3. However, under small scale conditions, the soluble protein expression levels of scTGP 14C2 and scTGP Z3 were relatively poor: as shown in Fig 1B the amounts of soluble protein obtained for 14C2 and Z3 were ~30% and ~16%, respectively, of those obtained with scTGP P1-1. Scaling up to 1 liter increased the absolute, but not the relative, amounts obtained (Table 1). Notwithstanding the relatively low expression levels of these scTGP proteins, they were far higher in all cases than the amounts of scFv produced, with conversion to scTGP improving protein production 3–12 fold, making these useful reagents for immunofluorescence as shown in Fig 1C, where both scTGP 14C2 and Z3 are able to specifically label the surface of cells expressing influenza M2 in a one-step assay. These results prompted us to conclude that while expression in the scTGP format generally increases expression levels, there remains a wide range in the absolute amounts of scTGPs that can be obtained, and high throughput screening of these chimeric proteins may allow rapid isolation of the best antibodies in this format.

### Development of a high throughput method for scTGP library construction, analysis, and enrichment

The combination of the inherent fluorescence and binding activity of scTGPs suggests the possibility of creating a high throughput flow cytometry screen that assesses both properties simultaneously, with the rationale that simultaneous selection for binding and fluorescence would allow the isolation of antibodies that retained binding, expression, and fluorescence in the scTGP format, which in turn would translate to easier expression and purification for these proteins. We constructed scTGP libraries derived from selection outputs obtained after two rounds of phage selection and two yeast sorts [14,15] on two human proteins: Cyclin-Dependent Kinase 2A (CDK2A), a human protein important in cancer biology, and Ubiquitin Specific Peptidase 11 (USP11), associated with bronchiectasis. As shown in Fig 2, plasmid DNA from these selection outputs provided input DNA for inverse PCR and subsequent CPEC assembly [40] with the TGP amplicon inserted as a linker between VL and VH. This technique provided a library of antibodies that were specific to the target of interest in the scTGP format. This library contained a mixture of scFvs, some functional as scTGP, and some that lost display and/or binding when converted to the scTGP format.

Using flow cytometry, we compared the expression of both the parental scFv outputs and the newly constructed scTGP ones. A comparison of the display level on yeast cells was

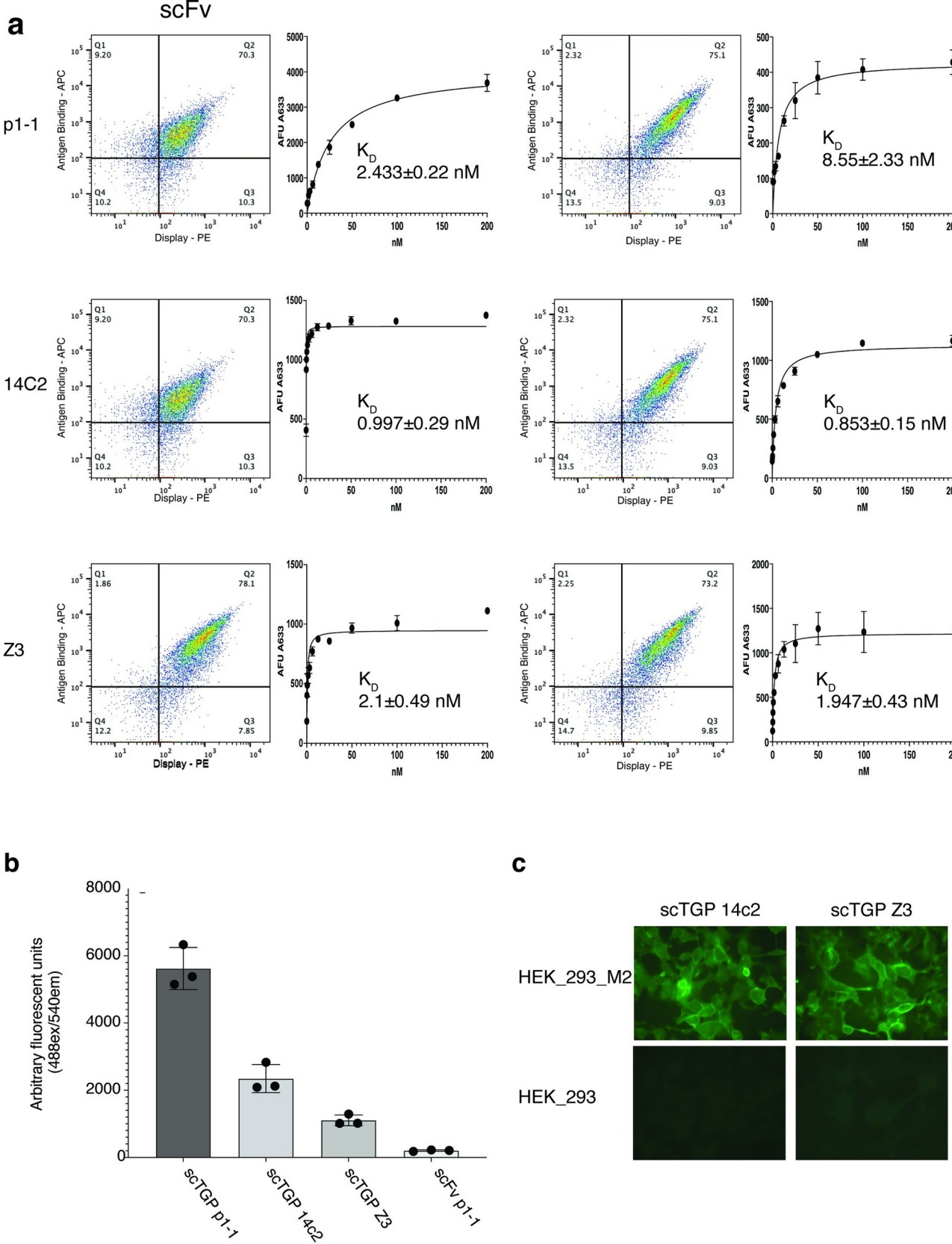

**Fig 1. Converting scFvs to scTGP format and their characterization.** The antibodies 14C2 and Z3 were originally isolated as IgGs and recognize the ecto domain of the M2 influenza protein. The antibody named p1-1, was isolated as an scFv in an *in vitro* selection campaign recognizes the phosphorylated ITAM peptide of FceR1 receptor of the allergy pathway. **a)** Yeast display plot for the three antibodies in both scFv and scTGP formats. The x-axis shows the expression level measured using PE labeled anti-SV5 antibody, which recognizes the SV5 expression tag in the vector system. The y-axis depicts the antigen recognition signal. Biotinylated M2 peptide was the specific antigen for the 14C2 and Z3 and biotinylated ITAM peptide p1-1 was the antigen used to characterize p1-1. The biotinylated peptides were labeled with streptavidin Alexa 633. The $K_D$ (nM) values, and flow cytometer titration curves, are shown. **b)** Soluble protein expression from 1mL protein expression culture. The green fluorescence of the POP culture supernatant is shown. Here scTGP and scFv p1-1 are used for comparison. **c)** ScTGPs were used as one-step fluorescent microscopy reagents. Mammalian cells constitutively expressing the M2 protein of influenza A (HEK M2) and control cells (HEK) were incubated with purified scTGPs 14C2 and Z3.

evaluated by measuring the fluorescent signal obtained from an antibody conjugated with PE recognizing the SV5 tag for the outputs expressed as scFvs and scTGPs. For the scTGPs the intrinsic fluorescence of the TGP in the chimeric protein was also measured. For the CDK2A outputs 75.6% display was detected for the scFvs and 50% for the scTGPs (using anti-SV5 PE conjugated antibody), when the display was measured using the intrinsic fluorescent signal of the chimeric proteins, for the scFvs displayed on yeast there was no significant signal, while 74.4% of display was detected for the scTGPs constructs displayed on yeast. For the USP11 outputs those values were 73.5% for the scFvs and 50.9% for scTGPs when the SV5 was used for the detection, while the GFP signal gave a small background for the scFvs and 71.3% display value for the scTGPs constructs (Fig 3).

The antigen recognition was detected using biotinylated CDK2A and USP11 proteins and streptavidin Alexa 633 as secondary reagent. Fig 4A shows that as scFvs 57.1% of the CDK2A selected population bound CDK2A and 66.7% for the corresponding USP11 population. When recloned as scTGPs, this dropped to 40.6% for the CDK2A, and 41.5% for the USP11 populations respectively when the display was detected using SV5-PE, and 40.6% (CDK2A) and 45.3% (USP11) when detected by their intrinsic fluorescence.

Cells expressing fluorescent scTGP clones binding the target were sorted, with gates set as shown in Fig 4A. The sorted population was further cultured and analyzed for binding specificity. As shown in Fig 4B, more than 75% of both sorted populations bound their respective targets, but not the other, at 100 nM target concentration, whether the anti-SV5-PE antibody or the intrinsic green fluorescence were used to detect display.

## Sequence based analyses of scFv and scTGP sort libraries

The scFv outputs and the scTGP outputs obtained after a final sort were analyzed by next generation sequencing (NGS) to assess the selection diversity and identify the most abundant clones. The CDK2A scFv output showed eight dominant clones, with the most abundant VH (HCDR3: CASQGFQGDAFDIW) at 59.6%. The same HCDR3 was also extremely dominant in the scTGP output (~87.3%), with the other seven clones identified as scFvs also present but at different abundances (Table 2). The second most abundant scFv HCDR3 (CARG-TEGWFDPW) at 14.3%, was significantly depleted in the scTGP population (to 0.4%). A single HCDR3 was found dominant (81%) both in the USP11 scFv output and in the scTGP one (96%). In the scFv output NGS analysis three clones were present with abundance values above 1%, but such clones were lost in the scTGP output, where four other clones were identified but with such low abundance values that they can be considered background (Table 3).

**Table 1. Comparison of protein purified from 1lit culture in scFv and scTGP formats.**

|       | p1-1     | Z3       | 14C2     |
|-------|----------|----------|----------|
| scFv  | 2.19 mg  | 0.44 mg  | 0.59 mg  |
| scTGP | 13.38 mg | 1.22 mg  | 5.84 mg  |

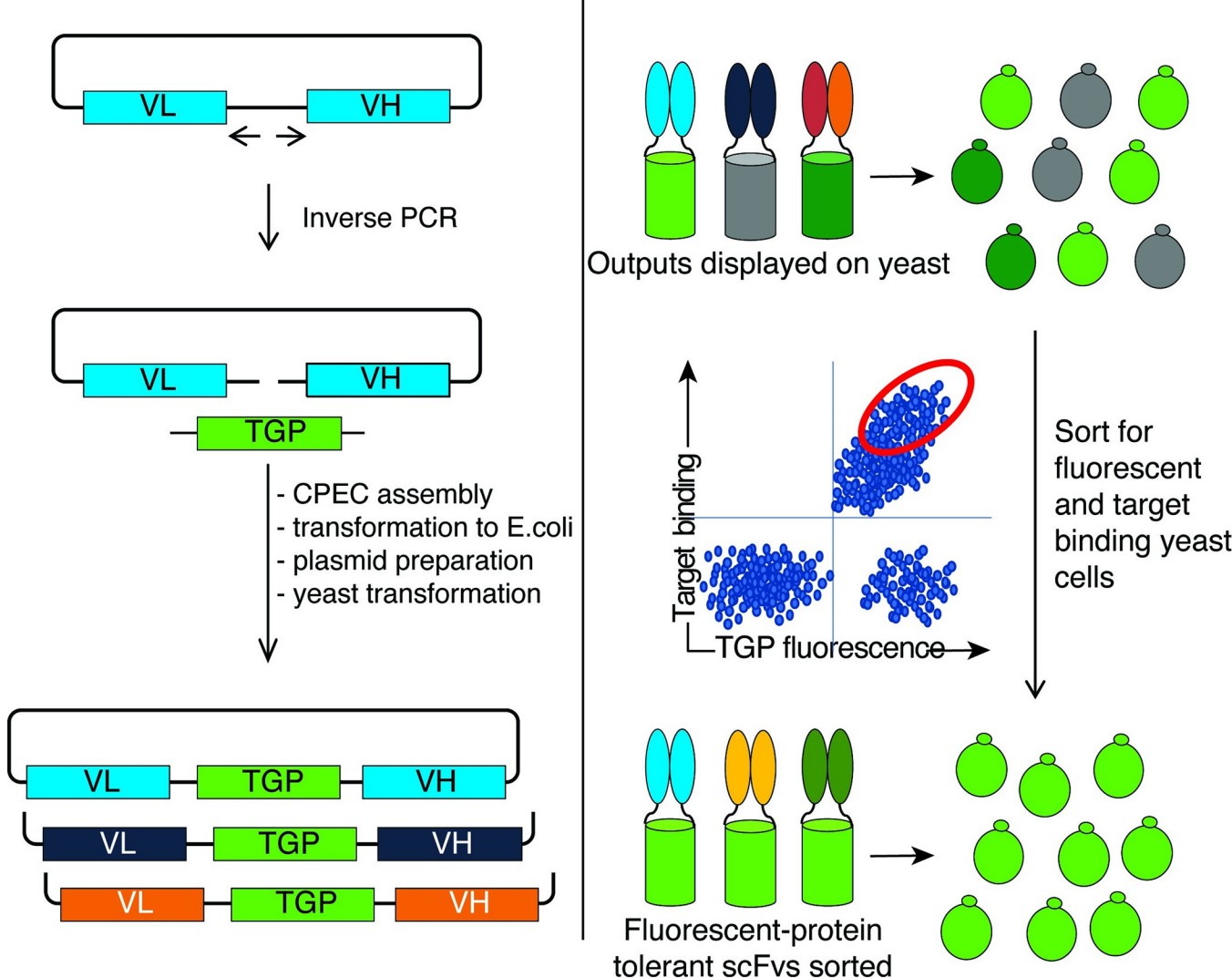

**Fig 2. Schematic depicting scFP library preparation and sorting to obtain antibodies tolerant to fluorescent protein insertion.** The selected scFv library is amplified by inverse PCR and CPEC assembly is used to build the scFP library. This library is displayed on yeast and sorted for antigen recognition and fluorescence. The sorted library is cloned into protein expression vectors for scFP production.

## Monoclonal analyses for selected scTGPs–protein expression and binding characteristics

Sanger sequencing of 96 clones from each of the scTGP outputs was used to identify unique scTGPs. From this limited sampling only six unique scTGPs (two for CDK2A and four for USP11) (Table 4) were identified. Interestingly, for the antibodies specific to CDK2A while the VH-CDR3s differed the VL-CDR3s were identical, while for the USP11 clones a single VH-CDR3 was identified, in agreement with its dominance seen in the NGS analysis, the VL-CDR3s were similar, differing by 1–2 amino acids.

These clones were chosen to evaluate the protein expression and binding characteristics of the scTGPs.

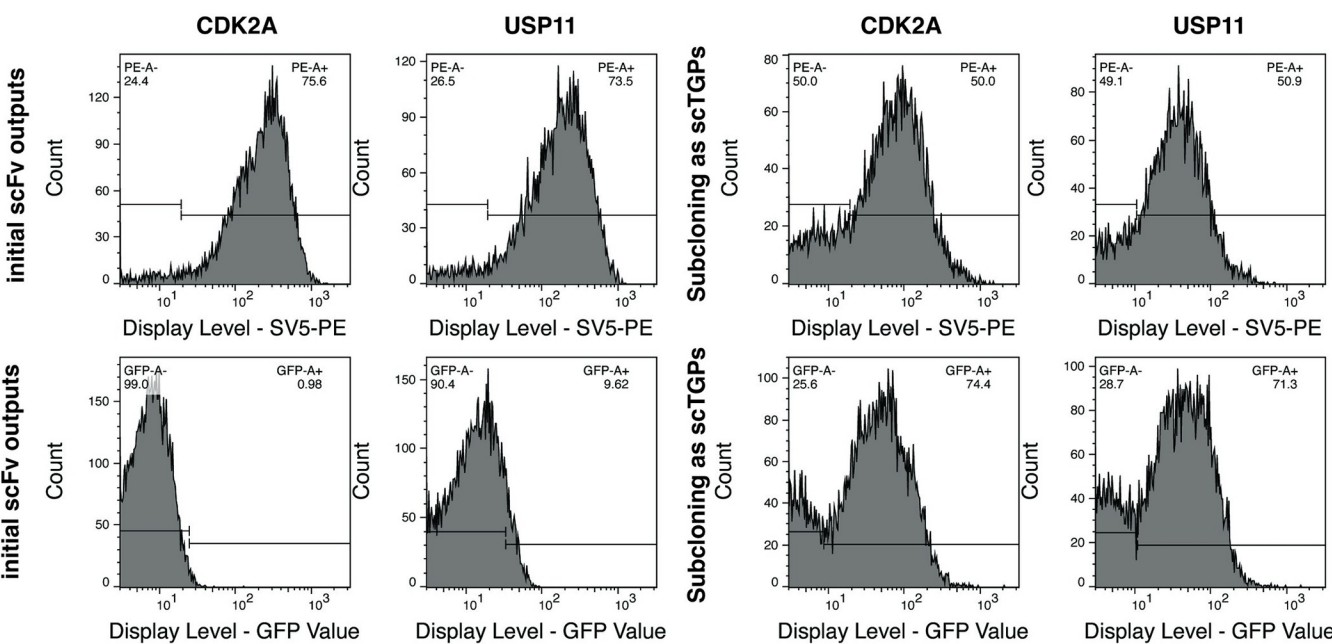

**Fig 3. Comparison of yeast display levels for the parent scFv library and the scTGP library.** The display levels shown on the x-axis were assessed with anti-SV5 labeled with PE (YG-PE) or the intrinsic green fluorescence of TGP. The left-hand panel provides data for the scFv library while the right-hand panel provides data for the scTGP libraries. Data for CDKA2A and USP11 libraries are given.

The expression of the identified clones was tested by expressing them in small scale and measuring the fluorescent values of the crude bacterial extracts, showing a significant signal for all clones (Fig 5A).

Binding characteristics of the six scTGPs against CDK2A and USP11 were evaluated using yeast display while protein expression, together with functionality, was tested by FLISA. These scTGPs clones were selected from the yeast display output and predictably, antigen recognition using this system, was confirmed for all six scTGPs (Fig 5B). However, compared to the flow cytometry results, the FLISA, performed with normalized protein concentrations, showed variability in the binding signal (Fig 5C). The scTGP CDK2A-A8 which contains the dominant VH-CDR3 in CDK2A selection showed better binding, while the CDK2A-F2 clone recognized the target with a weaker signal. The antigen recognition signals for the four anti-USP11 clones (USP11-B6, USP11-C3, USP11-G3, USP11-H6) were very similar, with more inconsistency in the FLISA than the flow cytometry. All four of these antibodies have identical VH-CDR3 and highly similar VL-CDR3, hence their similar function was not unexpected.

Another key result shown in Table 5 is the relatively high expression levels obtained for the best anti-CDK2A and anti-USP11 scTGPs that were selected as scTGPs compared to p1-1, Z3, and 14C2 scTGPs (Table 1) that did not go through yeast-based sorting. The amounts of purified protein obtained from some of these clones (18-25mg) are extremely high, highlighting the advantages of conversion to scTGPs. Interestingly, some of the other clones had expression values comparable to the scTGPs shown in Table 1.

Having demonstrated the value of using intrinsic fluorescence and binding activity to isolate well-expressed scTGPs, we were interested in assessing whether this reflected particularly well behaved scTGPs, or also applied to the derivative scFvs. Consequently, we assessed the expression levels of clones initially identified as scTGPs expressed as scFvs. The genes corresponding to two of the clones recognizing CDK2A (CDK2A-A8 and CDK2A-F2) and two

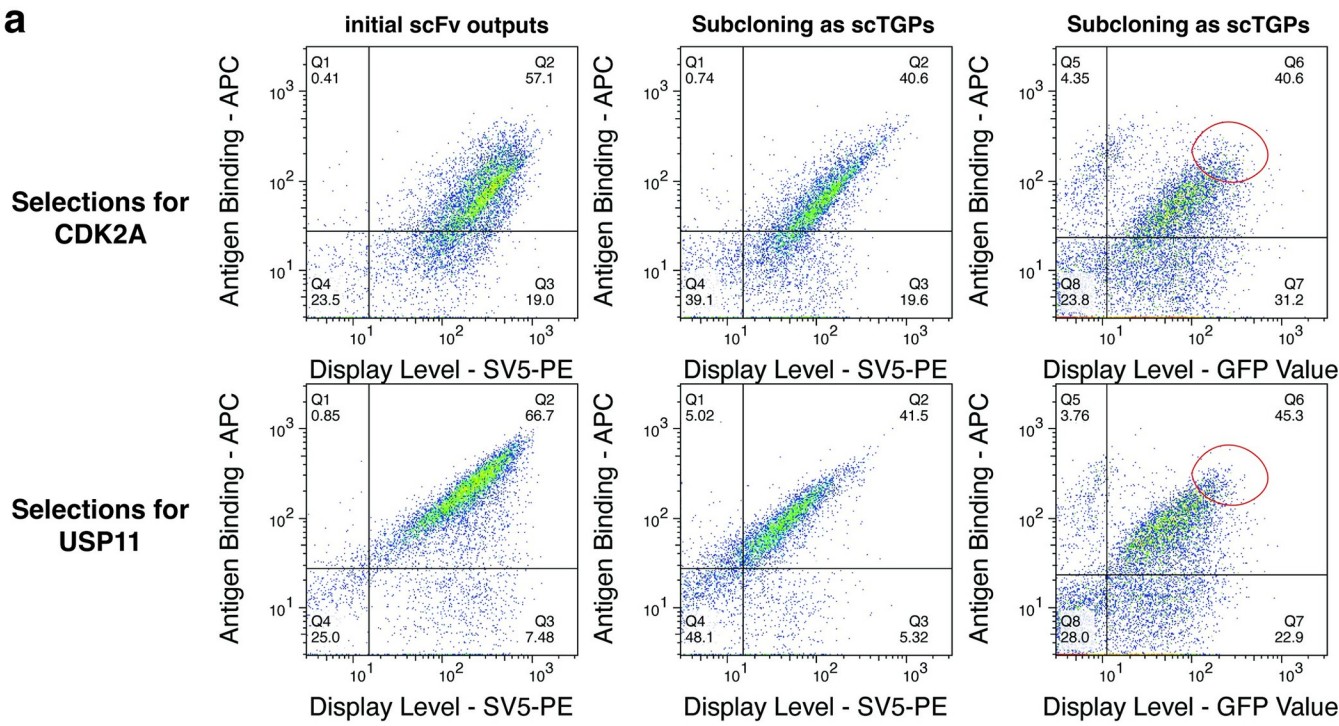

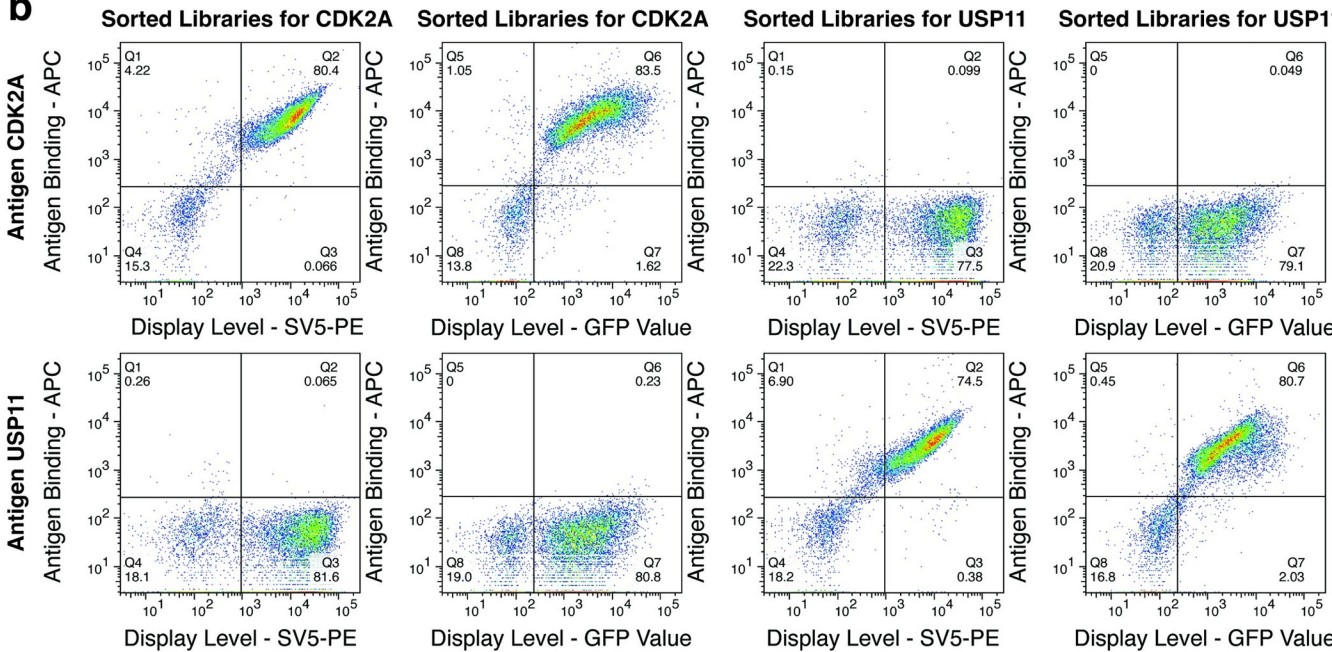

**Fig 4. scTGP libraries—comparison with scFv, sorting, and specificity. a)** the top row compares the display levels and antigen recognition of the parental scFv libraries and the corresponding scTGP libraries. The display levels shown on the x-axis were assessed with anti-SV5 labeled with PE (YG-PE) or the intrinsic green fluorescence of TGP. The PE signal is detected at 561nm excitation and 582/15nm emission and the TGP fluorescence is detected at 488nm excitation & 530/30nm emission. The y-axis depicts the antigen recognition signal obtained using biotinylated antigen plus streptavidin Alexa 633 at 635nm

excitation with 660/20nm emission. The right most section of panel A also shows the sorting gate used to select the best scTGP binders. **b)** the specificity of the sorted population is shown. The display is assessed by anti SV5 PE and green fluorescence, and the two antigens are used as specificity controls for one another. Data for CDK2A and USP11 libraries are provided.

recognizing USP11 (USP11-B6 and USP11-C3) were synthetized and expressed as scFvs (S1 Table for sequences). Somewhat surprisingly, when expressed as scFvs it was difficult to identify a single band at the expected molecular weight in polyacrylamide gel electrophoresis (Fig 6A), whereas scTGP expression revealed a prominent band after purification (Fig 6B). Size exclusion chromatography (SEC) demonstrated (Fig 6C–6E, S1 Fig) that scTGP expression was predominantly comprised of monomeric, non-aggregated molecules, further providing evidence for the high quality of the antibody reagents obtained when expressed as scTGP. Affinities of these purified clones were evaluated by SPR, and ranged from 4 to 160 4 nM (Fig 7).

## Discussion

Antibodies can recognize and bind specifically to many different target molecules including proteins, carbohydrates, lipids, and nucleic acids [7,13]. While immunization is the commonest method to generate antibodies, in vitro display methods provide opportunities to select for exquisitely specific properties, including distinguishing between proteins differing by a single surface amino acid [44], the generic tyrosine sulfate post-translational modification [9] and specific tyrosine phosphorylation sites [10]. Some of these antibodies have value as research reagents, with detection using traditional secondary antibodies. One advantage of in vitro display is the relative ease with which Fc domains can be changed within the context of modular vectors [45], allowing detection by different species-specific secondary reagents. The potential of in vitro selected antibodies can be further enhanced by fusion to functional polypeptides that provide new properties, such as fluorescence [19,30,31], multimerization [46] or enzymatic activity [5,47,48]. However, not all antibodies are equally tolerant to fusion to specific functional polypeptides, and this is particularly problematic where the functional polypeptide is normally expressed in a different cellular compartment to the antibody, as is the case for fluorescent proteins: fluorescent proteins are naturally expressed in the reducing environment of the cytoplasm, antibodies are naturally expressed in the oxidizing environment of the secretory pathway. While the concept of antibody-GFP fusions is extremely attractive, with a couple

**Table 2. VH-CDR3 amino acid sequences of scFv and scTGP clones that recognize CDK2A with relative abundance of each clone in the sorted yeast sub-library.** Identical CDRS in both libraries are indicated with same colors.

| CDK2A scFv output | Abundance (%) | CDK2A scTGP output | Abundance (%) |
|---|---|---|---|
| CASQGFQGDAFDIW | 59.55 | CASQGFQGDAFDIW | 87.31 |
| CARGTEGWFDPW | 14.34 | CARPLSGWYGDAFDIW | 5.71 |
| CARPLSGWYGDAFDIW | 1.83 | CARPYYGSGDAFDYW | 1.02 |
| CARVRGSGRVDYW | 1.50 | CATHSSGWYGDAFDIW | 0.98 |
| CARPYYGSGDAFDYW | 0.50 | CARVRGSGRVDYW | 0.65 |
| CATHSSGWYGDAFDIW | 0.48 | CASQGFQGDAFDYW | 0.46 |
| CARSWGADAFDIW | 0.23 | CARGTEGWFDPW | 0.42 |
| CASQGFQGDAFDYW | 0.20 | CARPYYGSGDAFDIW | 0.28 |
| | | CATHAAAGDYW | 0.20 |
| | | CARSWGADAFDIW | 0.12 |
| | | CASQGFQRDAFDIW | 0.10 |

**Table 3. VH-CDR3 amino acid sequences of scFv and scTGP clones that recognize USP11 with relative abundance of each clone in the sorted yeast sub-libraries.** Identical CDRS in both libraries are indicated with same colors.

| USP11 scFv output | Abundance (%) | USP11 scTGP output | Abundance (%) |
|---|---|---|---|
| CAREGGIGLSGWLDPW | 80.94 | CAREGGIGLSGWLDPW | 96.06 |
| CARDRAGDW | 2.32 | CARVRGSGRVDYW | 0.65 |
| CARDLTSWFDPW | 1.24 | CARGGSYGALDYW | 0.13 |
| CARDAHAFDIW | 1.02 | CAREGSIGLSGWLDPW | 0.10 |
| CAREGSIW | 0.94 | CAREGGIGLAGWLDPW | 0.10 |
| CARDARAFDIW | 0.78 | | |
| CARVPYGEGDFDYW | 0.73 | | |
| CARDPGVW | 0.56 | | |
| CARELGAGLYGPMDVW | 0.53 | | |
| CARDRYGMDVW | 0.50 | | |
| CARERGTMVRGWFDPW | 0.49 | | |
| CARERSRGYHRRFDPW | 0.44 | | |
| CARGWLRSGFDYW | 0.42 | | |
| CARDLDIW | 0.37 | | |
| CITGTGYYYYGMDVW | 0.36 | | |
| CARSMGATTPYGYFQHW | 0.29 | | |
| CARGPILGAFDIW | 0.27 | | |
| CARDGLAFDIW | 0.27 | | |
| CVRGIDFW | 0.25 | | |
| CARGVDVW | 0.25 | | |
| CARDLGPAWGFDYW | 0.24 | | |
| CARGGSYGALDYW | 0.20 | | |
| CARGVDIW | 0.20 | | |

of exceptions, poor expression levels have prevented widespread adoption. The exceptions include GFP fusions to scFv [22] or VHH [23] expressed in the cytoplasm using antibody fragments stable in the absence of disulfide bonds. The scTGP format described here, and in previous publications [30–32], in which the fluorescent protein acts as the linker between the VH and VL and can be expressed under both reducing and oxidizing conditions is another example. However, in both cases expression can be unpredictable, and depends upon inherent antibody properties. In the case of scTGPs, we have found many scFvs are not particularly well expressed in the scTGP format, and it is difficult to predict *a priori* without individual testing. If these formats are to become more popular, a high throughput method to identify suitable antibody candidates from a library of specific binders has the potential to significantly improve the technology.

**Table 4. VH and VH -CDR3 amino acid sequences of monoclonal scTGPs that recognize CDK2A and USP11.**

| Clone | VH-CDR3 | VL-CDR3 |
|---|---|---|
| CDK2A-A8 | CASQGFQGDAFDIW | YYCQVWDSSSDPYVFG |
| CDK2A-F2 | CARGTEGWFDPW | YYCQVWDSSSDPYVFG |
| USP11-B6 | CAREGGIGLSGWLDPW | YYCQQYDSWPLTFG |
| USP11-C3 | CAREGGIGLSGWLDPW | YYCQQYDDWPLTFG |
| USP11-G3 | CAREGGIGLSGWLDPW | YYCQQYNDWPLTFG |
| USP11-H6 | CAREGGIGLSGWLDPW | YYCQQYNNWPLTFG |

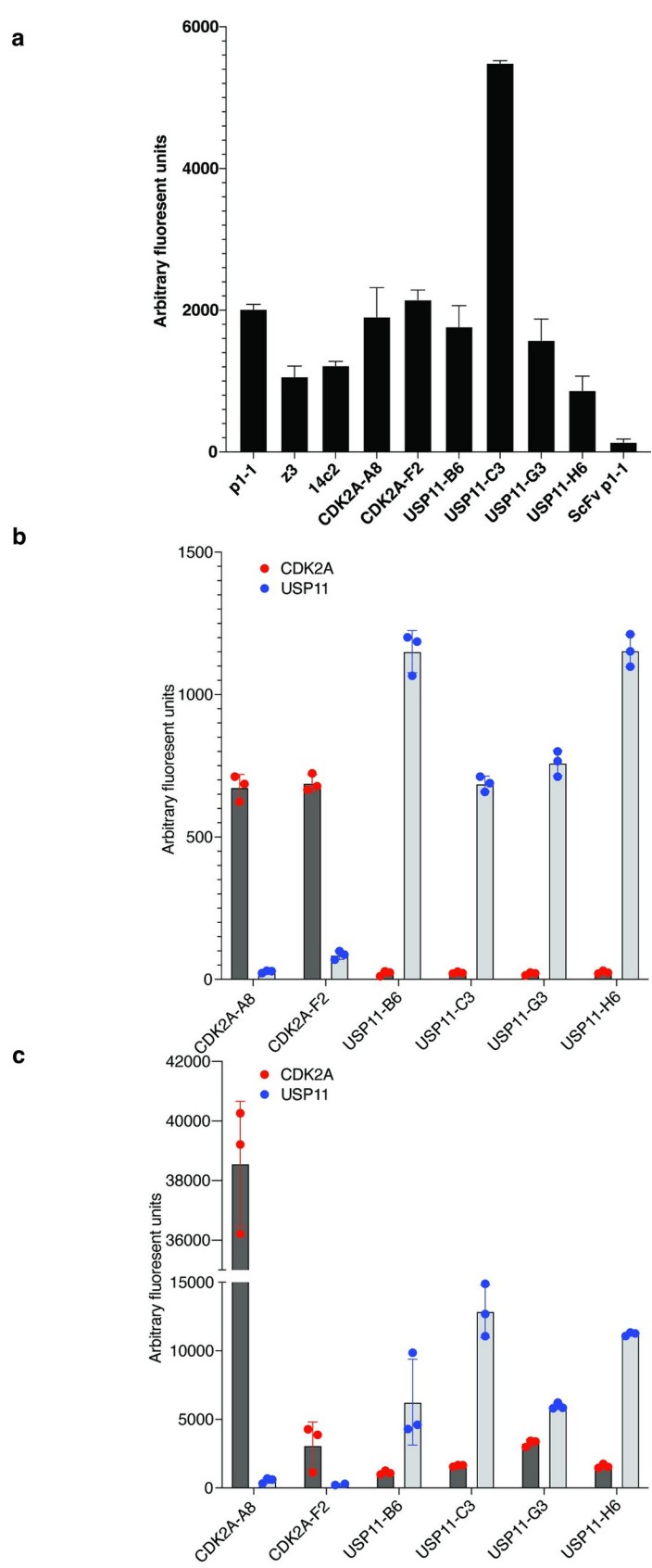

**Fig 5. Expression level comparison and binding characteristics of selected scTGPs. a)** data on *E.coli* based protein expression from various scTGP clones. The fluorescence signal obtained from soluble protein present in 100 ml of pop culture supernatant obtained from 2mL growth cultures are compared. The experiment was performed in triplicate and standard deviation is shown as error bars. Antigen binding data for the scTGPs displayed on yeast, **b**), and in the purified protein format using FLISA, **c**). The CDK2A and USP11 antigens are used as cross specificity controls for all six scTGPs (x-axis) and fluorescent signal obtained upon binding is shown on y-axis. The binding assays were performed in triplicate and the average and standard deviation (error bars) are shown.

Phage and yeast display technologies have been successfully used to isolate a vast array of specific antibodies in the past thirty years [7,49,50]. Phage display is a technique that isolates antibody fragments based on binding activity alone [51], while yeast display [52] identifies binders indirectly, usually staining yeast with two different fluorophores: one to assess the antibody display level with a fluorescent anti-tag antibody, and a second to label the target antigen, and hence the amount of bound target. By setting gates that consider both display levels and the amount of bound target, it is possible to isolate high affinity antibodies. As scTGPs are intrinsically fluorescent molecules, this raises the possibility of using the intrinsic fluorescence rather than fluorescent anti-tag antibodies to directly isolate scTGPs that bind target and are fluorescent. The data presented here shows the feasibility of using this high throughput approach to identify functional scTGPs. By carrying out selection for both specific binding and fluorescent properties from the yeast populations simultaneously, this should shorten the time required to produce such reagents. Working directly with binders in the scTGP format not only allows their direct enrichment as fluorescent high specific binders, but also ensures that selected clones have good production levels when expressed as recombinant proteins, in contrast to scFvs, which tend to show much lower expression levels, particularly when expressed in prokaryotic systems. Moreover, there is the risk that scFvs, when reformatted as scTGPs using tradition methods of recloning, may lose their binding activity.

Starting from a single-chain library [35] that was selected against two targets (USP11 and CDK2A) using a well-established phage and yeast display combination strategy [14,15], we obtained antigen-specific yeast populations that were used as inputs to create the scTGP sub libraries. CPEC assembly [40] was used to insert the TGP molecule between the two variable domains of the scFvs. By using the intrinsic fluorescence of the scTGPs to replace fluorescence normally mediated by the fluorescent anti-tag antibody, it was possible to sort a population that retained both fluorescence and binding activity. Moreover, when antibodies were tested as both scFvs and scTGPs it became clear that selection for scTGP properties (binding activity and fluorescence) did not translate into correspondingly favorable properties for the isolated scFvs, which did not show the same functional, non-aggregated, high expression levels.

Although we used thermal green protein (TGP) to create scTGPs, in part because of the extremely high stability of this protein, other fluorescent proteins (sfGFP, monomeric red, blue, cerulean and citrine fluorescent proteins) [30–32] have been previously used to create scFPs of different colors.

We have also successfully used superfolder GFP as the fluorescent protein for some antibodies [12], where we describe the development of antibodies specific to the cytoplasmic domain of the Influenza A M2 protein. As all other fluorescent proteins based on the beta barrel structure have structures like sfGFP and TGP, we do not foresee major issues in the

**Table 5. Purified scTGP proteins obtained from 1litre protein expression cultures.**

|  | CDK2A-A8 | CDK2A-F2 | USP11-B6 | USP11-C3 | USP11-G3 | USP11-H6 |
|---|---|---|---|---|---|---|
| **scTGP** | 2.34 mg | 25.1 mg | 18.57 mg | 19.04 mg | 4.28 mg | 1.47 mg |

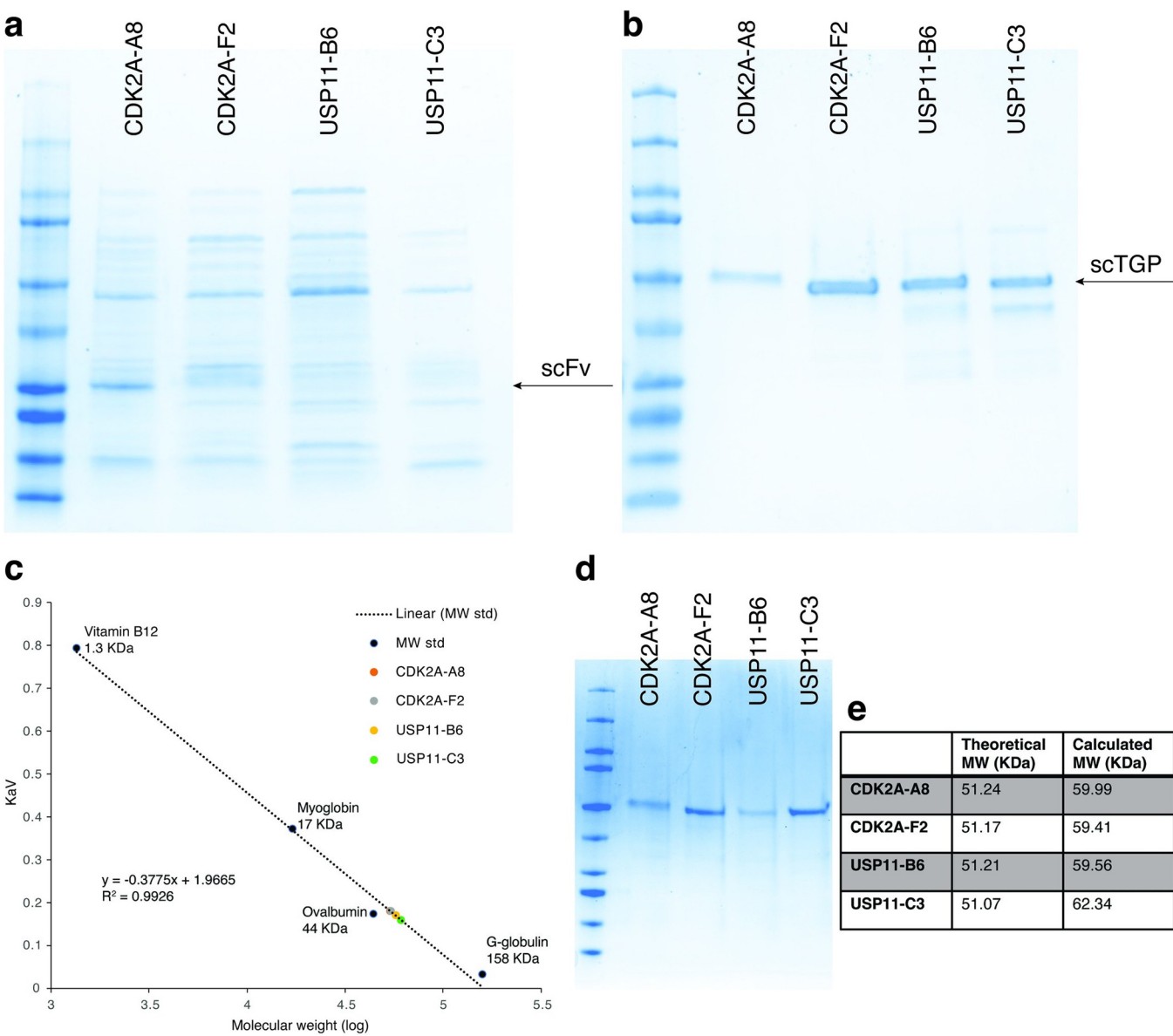

**Fig 6. Comparison of the quality of purified scFvs and scTGPs.** Panel **a)** shows scFvs and **b)** shows scTGPs purified by His tag affinity chromatography followed by TEV digestion. The expected molecular weights for scFvs are ~30 $K_D$ and for scTGPs ~51 $K_D$. **c)** Gel filtration chromatography results for the scTGP proteins. The proteins ran as monomers with calculated Mw 59–62 KDa based on standard calibration curve (theoretical molecular weight is ~ 51 KDa). The dotted line is the calibration curve calculated from the data for protein standards (R2 = 0.9926). The main scTGP peaks are shown as colored dots. **d)** SDS-PAGE shows protein samples ran as a single band **e)**. Comparison of theoretical and calculated MWs.

application in the system described in the present manuscript to other fluorescent proteins. The only possible caveat is that the creation of fluorescent antibody fusions, as described here, is likely to be more efficient with stable well-expressed monomeric fluorescent proteins. We anticipate the method we describe here would be similarly applicable to generating scFPs of any color.

Moreover, the methodology here described could be extended to proteins useable in other assays, such as enzymes. Of course, it would be necessary to test that their presence between the two variable regions of the antibodies would not have a steric negative effect. This would

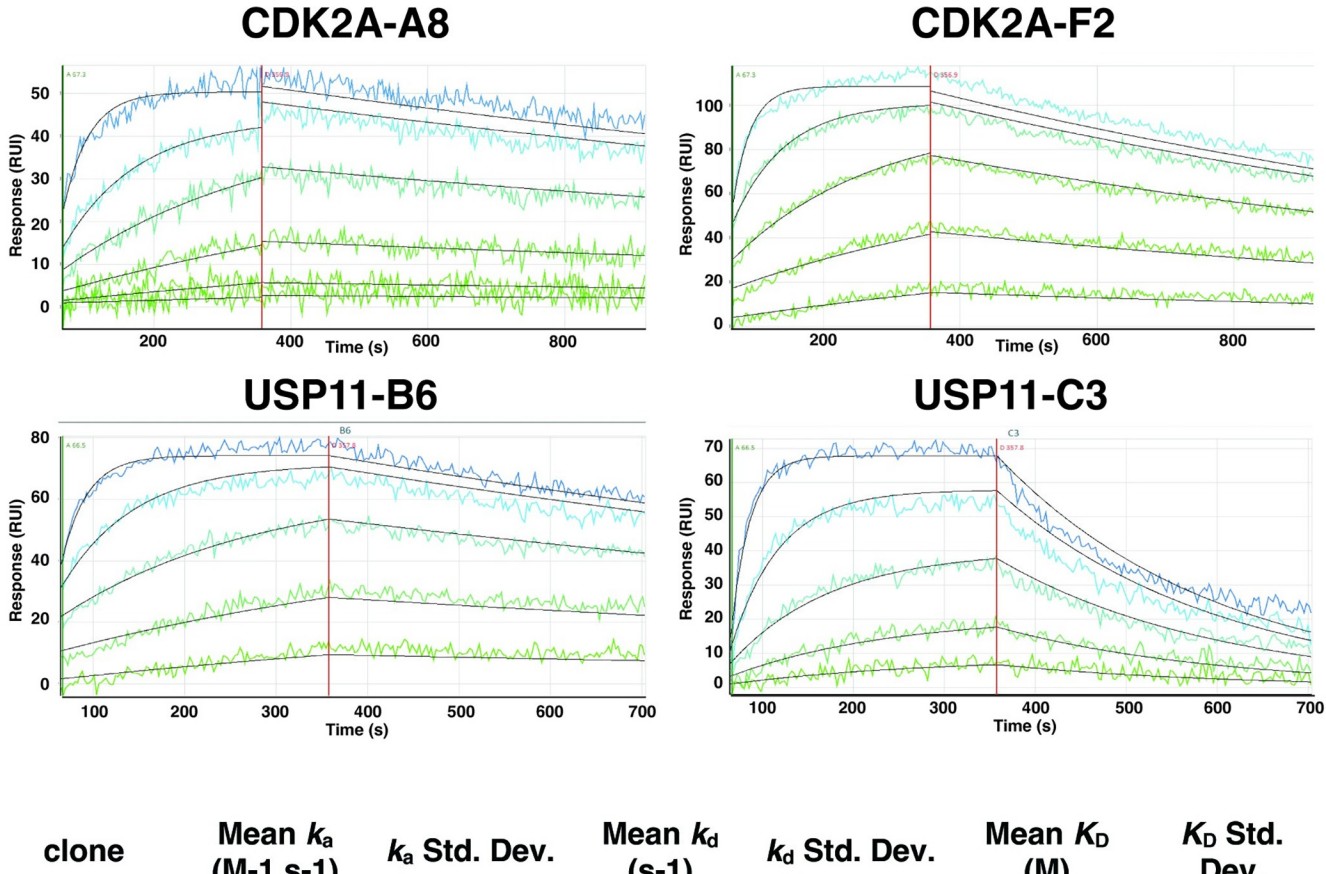

**Fig 7. SPR based affinity measurement for scTGPs.** The sensograms for four scTGPs (CDK2A-A8, CDK2A F2, USP11 B6 and USP11 C3) for their specific antigen is shown along with the association rate constant $k_a$ and dissociation rate constant $K_D$ as well as the affinity constant $K_D$ ($k_d$/ $k_a$).

| clone | Mean $k_a$ (M-1 s-1) | $k_a$ Std. Dev. | Mean $k_d$ (s-1) | $k_d$ Std. Dev. | Mean $K_D$ (M) | $K_D$ Std. Dev. |
|---|---|---|---|---|---|---|
| CDK2A-A8 | 4.1E+04 | 5.0E+03 | 3.2E-04 | 4.9E-05 | 8.0E-09 | 1.6E-09 |
| CDK2A-F2 | 1.9E+05 | 2.2E+04 | 6.8E-04 | 8.4E-05 | 3.7E-09 | 6.3E-10 |
| USP11-B6 | 1.9E+04 | 3.2E+03 | 8.9E-04 | 2.3E-04 | 4.8E-08 | 1.5E-08 |
| USP11-C3 | 2.3E+04 | 2.4E+03 | 3.8E-03 | 5.7E-04 | 1.6E-07 | 3.0E-08 |

likely be most effective if the N and C terminals of the inserted protein are close to one another, as is the case with fluorescent proteins.

Both CD2KA and USP11 selections were dominated by a single scFv clone. However, the use of next generation sequencing and the analyses described here, allowed the identification of additional less abundant clones that were tolerant to the use of TGP as a linker. Furthermore, it is possible that sorting yeast displaying different scFPs into distinct bins, reflecting their intrinsic fluorescence, followed by NGS could provide data suitable for machine learning with the potential to predict permissive antibody sequences.

The value of intrinsically fluorescent scTGP molecules lies in their ability to be used directly in one step binding assays in high throughput diagnostics and research. Here, we highlight the utility of scFPs in FLISAs [43] and fluorescent immunohistochemistry. However, there are many assays in which fluorescent secondary antibodies are presently used (e.g., FRET or flow cytometry) that could be easily replaced with scFPs if they could be easily derived. The wide

palette of available fluorescent proteins provides additional flexibility and complexity to the fluorescent colors with which antibodies can be labeled. The methods described here provide the means by which functional scFPs of many different colors could be easily derived.

## Supporting information

**S1 Fig. Chromatograms.** The chromatograms obtained for the four scTGPs are shown in comparison to the molecular weight marker.
(DOCX)

**S1 Table. DNA and amino acid sequences of the clones expressed and purified as scFvs and scTGP.**
(DOCX)

## Author Contributions

**Conceptualization:** Fortunato Ferrara, Andrew R. M. Bradbury.

**Funding acquisition:** Nileena Velappan, Andrew R. M. Bradbury.

**Investigation:** Nileena Velappan, Fortunato Ferrara, Sara D'Angelo, Devin Close, Leslie Naranjo, Madeline R. Bolding, Sarah C. Mozden, Camille B. Troup, Donna K. McCullough, Analyssa Gomez.

**Methodology:** Nileena Velappan, Devin Close, Camille B. Troup, Marijo Kedge.

**Supervision:** Nileena Velappan, Fortunato Ferrara, Sara D'Angelo, Andrew R. M. Bradbury.

**Visualization:** Nileena Velappan, Fortunato Ferrara.

**Writing – original draft:** Nileena Velappan, Fortunato Ferrara, Andrew R. M. Bradbury.

**Writing – review & editing:** Nileena Velappan, Fortunato Ferrara, Sara D'Angelo, Andrew R. M. Bradbury.

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
