## [Decision Letter · Decision Letter 0]

19 Jul 2022

PONE-D-22-17094Selecting directly for antibody fusion functionality by yeast displayPLOS ONE

Dear Dr. Andrew Bradbury

Thank you for submitting your manuscript to PLOS ONE. After careful consideration, we feel that it has merit but does not fully meet PLOS ONE’s publication criteria as it currently stands. Therefore, we invite you to submit a revised version of the manuscript that addresses the points raised during the review process.

The approach for the chimeric construct, scTGFP in this manuscript is quite novel and the authors used scFVs against two cell targets, CDK2A and USP11 as templates to construct the chimeric constructs. However, the MS needs to revised before publishing in PLOS One. I agree with the reviewers that the MS is novel and well written and results are well presented. The comments from reviewers are fair assessments and the responses for those comments will strengthen the quality of the MS. Thus, I highly recommend that the authors should revise the MS accordingly and provide the extra data assessment if feasible. Thank you for submitting your work in PLOS One.

We look forward to receiving your revised manuscript.

Kind regards,

Jinny L Liu, PhD

Academic Editor

PLOS ONE

Journal Requirements:

2.Thank you for stating the following in the Financial Disclosure section: 

"This work was supported by National Institutes of Health Grant P50GM085273 Foundation for the National Institutes of Health [P50GM085273]; Los Alamos National Laboratory's Laboratory Directed Research & Development grant 20220108ER."

We note that one or more of the authors are employed by a commercial company: Specifica Inc.

Reviewers' comments:

Reviewer's Responses to Questions

**Comments to the Author**

1. Is the manuscript technically sound, and do the data support the conclusions?

Reviewer #1: Partly

Reviewer #2: Yes

2. Has the statistical analysis been performed appropriately and rigorously? 

Reviewer #1: N/A

Reviewer #2: Yes

3. Have the authors made all data underlying the findings in their manuscript fully available?

Reviewer #1: Yes

Reviewer #2: Yes

4. Is the manuscript presented in an intelligible fashion and written in standard English?

Reviewer #1: Yes

Reviewer #2: Yes

5. Review Comments to the Author

Reviewer #1: Velappan et al PLOS ONE 2022

The manuscript is well written and presents an interesting and novel advancement of the group’s previous work exploring antibody FP fusions. A couple of additional experiments are suggested that should improve the manuscript quality and strengthen their conclusions.

1. The main message here is that screening of libraries of fusions works better than screening of scFvs alone followed by fusion. I am not sure the data fully support this conclusion. Comparing with a handful of other scFvs against different targets isn’t really that helpful. It would be more convincing if scFvs we selected against the same target and the authors find that a lot of them don’t work in this format. The NGS data essentially show they are getting the same result from either library selection.

2. Can the authors make the scFv versions of the new scTGPs they identified? Do those have similar expression levels? Comparing expression yields of new scTGPs (Table 5) to different scTGPs made after scFv selection (Table 1) isn’t really the strongest comparator.

3. Any reason why the authors did not try the library screening with the M2 and ITAM antigens to obtain a direct correlation with these three antibodies 14C2, Z3, and p1-1?

4. The authors need to show SEC profiles of some of the purified fusions. Profiles of the scFvs and scTGPs in Fig 1, and at minimum the newly identified scTGPs in Fig 5, should be shown. The high purification yields in Table 5 are encouraging, but if these contain high % of aggregates they might not be as useful and this could certainly impact/inflate binding affinities.

5. It is not clear how the kds for p1-1, Z3 and 14C2 scFv and scTGPs were determined in Fig 1A. Please add more detail to the methods/results. In addition, please show the data (ie, flow titration curves or ELISA curves) in the figure (along with SEC profiles).

6. In addition to scTGP binding by flow and FLISA (Fig 5), affinity determination (SPR, BLI) of the new scTGPs is also suggested since the authors have pure biotinylated Ag and purified scTGPs on hand. Additional specificity controls could be immobilized and evaluated here too. If aggregates are found in the SEC profiles, monomer fractions should be used for affinities/kinetics.

7. Are the scTGP proteins used for the FLISA in Fig 5 purified?

8. At first read, the manuscript title sounds awkward. Consider revising to “Direct selection for functional antibody-fusions by yeast display” or similar.

9. “USP11” is missing in the title of Table 4

Reviewer #2: The authors share a novel method to select antibodies that can be used to develop fluorescent chimera. This technique builds on their previously published method that uses split GFP for fluorescent labeling of antibody fragments. For this paper, thermal green protein is cloned into the linker of VH and VL of scFv fragments and then yeast display flow cytometry is used to select for both binding and fluorescence, which is more efficient than individual testing. The technique was validated by evaluation of activity of selected chimeras against USP11 and CDK2A.

Selection of ab fragments for binding, including by yeast display, has been the subject of much investigation. The technique presented in this paper logically builds on this foundation to incorporate selection for fluorescent function. Chimeras optimized for binding and fluorescence can be identified more easily than by other strategies.

Here are a few specific comments:

Introduction, 2nd paragraph, bottom of page – consider clarifying what “expressed well” means. Can this be quantified?

Methods / Results – would consider including dose response curves that include physiologic levels of CDK2A and USP11. It would be helpful to show that the chimeras are useful at these levels.

Methods, Fluorescence microscopy – Explain why the HEK293 cell line was transfected with M2 protein. Reference could be made to binding specificity of the antibodies used in this study.

For discussion – Table 5 shows expression of best scTGPs that did not go through yeast-based sorting. These might be missed by the yeast display flow strategy. How might this challenge be addressed while maintaining high throughput?

Discussion – consider mentioning how much time would be spent on chimera selection with the yeast based flow technique vs traditional methods.

Has the group attempted this method with other fluorescent proteins? Would insertion site, steric or other requirements affect binding differently compared to GFP? Are there size limitations on the insert?

Would this technique be useful with inserts with other assayable functions?

Minor editing comments:

Current figures are very low resolution. I could not see the signal intensities.

Define acronyms only at first use. For example, PE seems to be defined twice.

6. PLOS authors have the option to publish the peer review history of their article (what does this mean?). If published, this will include your full peer review and any attached files.

Reviewer #1: No

Reviewer #2: **Yes: **Chuen-Yen Lau

---

## [Author Response · Author response to Decision Letter 0]

5 Jan 2023

Point-by-point response to the reviewers’ comments.

Reviewer #1

The manuscript is well written and presents an interesting and novel advancement of the group’s previous work exploring antibody FP fusions. A couple of additional experiments are suggested that should improve the manuscript quality and strengthen their conclusions.

We appreciate that the reviewer found our manuscript a step forward over our previous published work and we agree that the additional suggested experiments improved the general strength of our findings.

1. The main message here is that screening of libraries of fusions works better than screening of scFvs alone followed by fusion. I am not sure the data fully support this conclusion. Comparing with a handful of other scFvs against different targets isn’t really that helpful. It would be more convincing if scFvs we selected against the same target and the authors find that a lot of them don’t work in this format. The NGS data essentially show they are getting the same result from either library selection.

We appreciate the reviewer’s comment. As the reviewer stated, the main goal is to show that the described process speeds up the possibility of obtaining well-expressed fluorescently “labeled” antibodies. The NGS data show that the selections were heavily skewed toward a few binders, but this is mostly due to the performance of the antibody library against those specific targets. We think the manuscript shows (especially after performing the extra experiments suggested by the reviewer, see below) that in the end even poorly expressing scFvs improve their expression when fused to TGP, as a linker between VL and VH, providing valuable tools without lengthy subcloning and optimization.

2. Can the authors make the scFv versions of the new scTGPs they identified? Do those have similar expression levels? Comparing expression yields of new scTGPs (Table 5) to different scTGPs made after scFv selection (Table 1) isn’t really the strongest comparator.

We thank the reviewer for this suggestion. We synthesized genes for 4 scFvs (CDK2A- A8, F2 and USP11 – B6, C3) and compared them to corresponding scTGPs. The new data shown in Figures 6A and 6B shows a substantial improvement in quantity and quality of purified protein for the scTGPs. 

3. Any reason why the authors did not try the library screening with the M2 and ITAM antigens to obtain a direct correlation with these three antibodies 14C2, Z3, and p1-1?

14C2 and Z3 antibodies were originally identified as IgGs and we converted to scFv and scTGP format by gene synthesis. Hence, an scFv selection output was not available for conversion to library format. For the p1-1 antibody, it could have been possible to convert the original phage selection library to the scTGP library format and perform similar experiments to those carried out for CDK2A and USP11. However, limited time and funding resources prevented us from carrying out these experiments.

4. The authors need to show SEC profiles of some of the purified fusions. Profiles of the scFvs and scTGPs in Fig 1, and at minimum the newly identified scTGPs in Fig 5, should be shown. The high purification yields in Table 5 are encouraging, but if these contain high % of aggregates they might not be as useful and this could certainly impact/inflate binding affinities.

We thank the reviewer for this suggestion. The size exclusion chromatography (SEC) analysis showed that the scTGPs elute have strong monomer peaks and SDS PAGE followed by Coomassie staining shows a single band (Figure 6C and Figure 6D). This provides further evidence for the high quality and quantity antibody reagents obtained when the scTGP format is used. We generated an extra figure summarizing these new results and have described them in the text.

5. It is not clear how the kds for p1-1, Z3 and 14C2 scFv and scTGPs were determined in Fig 1A. Please add more detail to the methods/results. In addition, please show the data (ie, flow titration curves or ELISA curves) in the figure (along with SEC profiles).

We measured the Kd for p1-1, Z3 and 14C2 using yeast displayed scFv and biotinylated peptide as antigen. We have amended the materials and method section to include more details. We also included the flow titration curves in Figure 1. Expression profiles for Z3 scTGP and 14C2 scTGP are very poor and not suitable for SEC. The scTGP p1-1 expresses very well and can be purified by SEC column and crystalized as shown in our previous manuscript (DOI: 10.1093/protein/gzaa029 ).

6. In addition to scTGP binding by flow and FLISA (Fig 5), affinity determination (SPR, BLI) of the new scTGPs is also suggested since the authors have pure biotinylated Ag and purified scTGPs on hand. Additional specificity controls could be immobilized and evaluated here too. If aggregates are found in the SEC profiles, monomer fractions should be used for affinities/kinetics.

We agree with the reviewer that nowadays a validation of purified antibodies or antibody derivates needs to be conducted with more refined methods. For this reason, we used the four purified scTGP molecules in a SPR experiment to have a more precise affinity measurement (shown in Figure 7).

7. Are the scTGP proteins used for the FLISA in Fig 5 purified?

Yes, these proteins were purified with Talon resin followed by TEV digestion, which gives a single band on SDS PAGE gel.

8. At first read, the manuscript title sounds awkward. Consider revising to “Direct selection for functional antibody-fusions by yeast display” or similar.

Title changed to:

Direct selection of functional fluorescent-protein antibody fusions by yeast display

9. “USP11” is missing in the title of Table 4

Thank you for catching this mistake, we have fixed the error

Reviewer #2: The authors share a novel method to select antibodies that can be used to develop fluorescent chimera. This technique builds on their previously published method that uses split GFP for fluorescent labeling of antibody fragments. For this paper, thermal green protein is cloned into the linker of VH and VL of scFv fragments and then yeast display flow cytometry is used to select for both binding and fluorescence, which is more efficient than individual testing. The technique was validated by evaluation of activity of selected chimeras against USP11 and CDK2A.

Selection of ab fragments for binding, including by yeast display, has been the subject of much investigation. The technique presented in this paper logically builds on this foundation to incorporate selection for fluorescent function. Chimeras optimized for binding and fluorescence can be identified more easily than by other strategies.

Here are a few specific comments:

? Introduction, 2nd paragraph, bottom of page – consider clarifying what “expressed well” means. Can this be quantified?

This is a general comment reflecting the finding that there is usually a correlation between expression and display 

? Methods / Results – would consider including dose response curves that include physiologic levels of CDK2A and USP11. It would be helpful to show that the chimeras are useful at these levels.

We were not able to find precise physiologic levels of the two antigens, however we think that the dose response curves (SPR profile) cover physiological levels.

? Methods, Fluorescence microscopy – Explain why the HEK293 cell line was transfected with M2 protein. Reference could be made to binding specificity of the antibodies used in this study.

We thank the reviewer for this discussion point. The HEK M2 cell lines were originally prepared for characterization of the h14C2 antibody described in the publication by Gabbard et al (doi: 10.1093/protein/gzn070). We subsequently also used these cell lines in our autophagy project and related publications (doi: 10.1111/mmi.14865 , DOI: 10.1080/19420862.2020.1843754 ) Gabbard et al. and Velappan et al. are reference in the current manuscript.

? For discussion – Table 5 shows expression of best scTGPs that did not go through yeast-based sorting. These might be missed by the yeast display flow strategy. How might this challenge be addressed while maintaining high throughput?

What we meant when discussing the results in Table 5 is that the scTGP clones were indeed obtained after the yeast-based sorting, and their expression level is higher compared to the first antibodies characterized as scTGPs derived from previous selection campaign where the yeast display scTGP-based approach was not used and whose expression is shown in Table 1. 

? Discussion – consider mentioning how much time would be spent on chimera selection with the yeast based flow technique vs traditional methods.

We thank the reviewer for the interesting consideration and have added a paragraph in the Discussion Section.

? Has the group attempted this method with other fluorescent proteins? Would insertion site, steric or other requirements affect binding differently compared to GFP? Are there size limitations on the insert?

In addition to the TGP used in the preset manuscript, we have successfully used superfolder GFP as the fluorescent protein for some antibodies, as described in Velappan et al. (DOI: 10.1080/19420862.2020.1843754 ), where we describe the development of antibodies specific to the cytoplasmic domain of the Influenza A M2 protein. As all other fluorescent proteins based on the beta barrel structure have structures similar to sfGFP and TGP, we do not foresee major issues in the application in the system described in the present manuscript to other fluorescent proteins. The only possible caveat is that the creation of fluorescent antibody fusions, as described here, is likely to be more efficient with stable well-expressed monomeric fluorescent proteins. We added these considerations in the discussion section.

? Would this technique be useful with inserts with other assayable functions?

Yes. In theory, we believe the methodology described could be extended to proteins useable in other assays, such as enzymes. Of course, it would be necessary to test that their presence between the two variable regions would not have a steric negative effect. This would likely be most effective if the N and C terminals of the inserted protein are close to one another, as is the case with fluorescent proteins. We added these considerations in the discussion section.

Minor editing comments:

? Current figures are very low resolution. I could not see the signal intensities.

We have uploaded the high resolutions figures to journal website

? Define acronyms only at first use. For example, PE seems to be defined twice.

We removed the second definition for PE

---

## [Editor Report · Decision Letter 1]

12 Jan 2023

Direct selection of functional fluorescent-protein antibody fusions by yeast display

PONE-D-22-17094R1

Dear Dr. Bradbury,

We’re pleased to inform you that your manuscript has been judged scientifically suitable for publication and will be formally accepted for publication once it meets all outstanding technical requirements.

Kind regards,

Jinny L Liu, PhD

Academic Editor

PLOS ONE
---

## [Editor Report · Acceptance letter]

16 Feb 2023

PONE-D-22-17094R1 

Direct selection of functional fluorescent-protein antibody fusions by yeast display 

Dear Dr. Bradbury:

I'm pleased to inform you that your manuscript has been deemed suitable for publication in PLOS ONE. Congratulations! Your manuscript is now with our production department. 

Kind regards, 

on behalf of

Dr. Jinny L Liu 

Academic Editor

PLOS ONE